# COVID-19 among Czech Dentistry Students: Higher Vaccination and Lower Prevalence Compared to General Population Counterparts

**DOI:** 10.3390/vaccines10111927

**Published:** 2022-11-14

**Authors:** Jan Schmidt, Lenka Vavrickova, Christos Micopulos, Jakub Suchanek, Nela Pilbauerova, Vojtech Perina, Martin Kapitan

**Affiliations:** 1Department of Dentistry, Charles University, Faculty of Medicine in Hradec Kralove, and University Hospital Hradec Kralove, 500 05 Hradec Kralove, Czech Republic; 2Department of Stomatology, Charles University, Faculty of Medicine in Pilsen, and University Hospital Pilsen, 323 00 Pilsen, Czech Republic; 3Department of Oral and Maxillofacial Surgery, Masaryk University, Faculty of Medicine, and University Hospital Brno, 625 00 Brno, Czech Republic

**Keywords:** COVID-19, vaccination, prevalence, dentistry, students

## Abstract

The restrictions on medical students’ clinical education during the COVID-19 pandemic has affected their professional readiness and often lengthened their training. These negative impacts are often considered a necessary price as clinical education is hypothesized to be associated with a high risk of pandemic spread. This work assesses this hypothesis based on COVID-19 epidemiological data among Czech dentistry students and their comparison to data of the Czech general population of similar age. We addressed two of the five Czech medical faculties (Charles University, Faculty of Medicine in Hradec Kralove and in Pilsen) providing dentistry study program with a survey. A total of 240 students participated, representing a 66.9% response rate. Over 75% of respondents participated in clinical education during the pandemic. The school environment was identified as a place of infection by only 9.8% of respondents who were aware of where they were infected. Overall, 100% of students used FFP2 respirators, and 75.3% used face shields or protective glasses while working with patients. By the end of May 2022, COVID-19 full vaccination and 1st booster rates among students were 93.8% and 54.6%, respectively, which is significantly higher (*p* < 0.0001, OR 7.3, 95% CI 4.4–12; *p* < 0.0001, OR 3.7, 95% CI 2.9–4.8, respectively) compared to their peers from the general population (67.1% and 24.4%, respectively). A total of 75.4% of respondents supported mandatory COVID-19 vaccination for healthcare professionals. To the same date, PCR and/or antigen test verified COVID-19 prevalence among students was 37.1%, while among peers from the general population, it was 45.1% (*p* = 0.015, OR 1.2, 95% CI 1.0–1.5). The combination of extensive protective measures and high vaccination against COVID-19 led to significantly lower COVID-19 prevalence among the students compared to their general population counterparts.

## 1. Introduction

The COVID-19 pandemic has brought significant changes in human society, including the operativeness of otherwise very stable sectors such as healthcare and education [1]. Due to the high risk of infection, the operation of medical facilities, including teaching at university hospitals, was restricted. These interventions have a negative impact on the availability of current and future health care. The curtailment of medical students’ education has affected their professional readiness and often lengthened their training [2,3]. The approach to this issue varied across countries, and different policies were applied. Some countries chose a very restrictive approach, while others maintained highly operative medical and education facilities. Opinions on which approach was correct differ. To assess them, it is necessary to analyze their impact from an educational and epidemiological point of view.

Numerous studies deal with the impact of restrictions on the education of medical students, confirming the negative effect on knowledge and skills [1,2,3,4]. These negative impacts are often considered a necessary price as clinical education is hypothesized to be associated with a high risk of pandemic spread. However, clinical teaching is a very specific field of education with higher infection risk but also with high hygienic and protective standards that medical students must be well versed in. It is unclear which of these factors prevailed, as studies on COVID-19 prevalence among medical students involved in clinical teaching are surprisingly scarce. Therefore, the above-mentioned hypothesis should be assessed. One option to address it is to measure the course of the pandemic among medical students who had frequent contact with patients in a high-risk environment during the pandemic and compare the results with the course of the pandemic in the general population of a similar age. Such a comparison provides the possibility to evaluate whether medical teaching is associated with a higher risk of infection with COVID-19. If so, it would confirm the legitimacy of study restrictions. If not, then this conclusion should be taken into account in the next waves of the pandemic and could contribute to future strategic decisions on limiting medical teaching. However, to the best of our knowledge, no studies are providing such an evaluation.

One of the medical fields associated with a high risk of COVID-19 infection is dentistry. Dental professionals work in close contact with patients, and a large number of droplets is formed during their work [5]. The same applies to dental students involved in clinical classes. Working with patients is a necessary and frequent part of dentistry study programs, and dental students are burdened with a high risk of infection than their graduated counterparts. This makes dental students a suitable group meeting the high infection risk criterion. An additional criterion is to identify a country highly affected by COVID-19 in which the clinical teaching of dental students was limited as little as possible.

Dentistry in the Czech Republic remained almost fully operational during the whole pandemic, although the Czech Republic was the country with the 7th highest prevalence of COVID-19 in the world during 2020–2021 [6,7]. Albeit the pandemic disrupted the teaching schedule of Czech medical faculties, the Faculty of Medicine in Hradec Kralove and the Faculty of Medicine in Pilsen carried out the vast majority of the dental clinical teaching. Additionally, the Czech government highly motivated healthcare professionals, including healthcare students, to vaccinate against COVID-19 and use protective equipment. Additionally, the Czech Republic is one of the countries with a high willingness of the population to get vaccinated, and this has been the case since the beginning of the pandemic. Multiple studies have shown that the Czech general population, health professionals and students have a positive attitude towards vaccination against COVID-19 [8,9,10,11,12]. Such conditions are optimal for assessing whether the high level of protective measures and vaccination outweighed the high risk of infection.

The aim of the study is to provide data on COVID-19 prevalence and vaccination of Czech dental students and its comparison with related data of their Czech general population counterparts.

## 2. Materials and Methods

### 2.1. Design

This study was designed as a bicentric self-administered, cross-sectional survey among 2nd–5th year dentistry students at Charles University, Faculty of Medicine in Hradec Kralove, and Faculty of Medicine in Pilsen (the academic year 2021/2022). These schools represent two of five faculties in the Czech Republic providing dentistry study program. The invitation to participate in the study and a link to the online questionnaire were sent to student official e-mail addresses together with information about the purpose of the study. The questionnaire was anonymous and did not include any identifying personal information. No remuneration or direct benefits were provided for participation. The survey was approved by the Ethics Committee of the University Hospital Hradec Kralove (approval number: 202204P07, approved date: 11 April 2022). The study was conducted and reported according to the Strengthening the Reporting of Observational Studies in Epidemiology (STROBE) guidelines for cross-sectional studies [13]. The STROBE checklist is provided as a Appendix A.

### 2.2. Instrument

The questionnaire was divided into three content sections. The first contained a combination of open and closed answers regarding demographic data (e.g., age, gender, year of study, place of study). In the second section, respondents were asked about their infection with COVID-19 (e.g., questions about how many times they were infected, when they were infected, what complications they had, how the disease was diagnosed). This section contained closed and semi-closed questions (additional comments were possible for some answers). In the third section, respondents were asked about their COVID-19 vaccination status (e.g., how many doses of the vaccine they received, when they received the doses, what types of vaccines were used, what the complications were). This section contained closed and semi-closed questions (additional comments were possible for some answers). Although the answers to the questions asked were mandatory, not every respondent was asked all questions as the questionnaire was adaptable (e.g., respondents who were not infected were not asked about complications associated with the infection). The questionnaire was developed in cooperation with experts from the Czech Dental Chamber, the academic community, and students of dentistry. The preliminary version of the questionnaire was consulted with representatives from among the respondents and adapted accordingly before proceeding to the reliability testing stage. The test reliability was evaluated through a group of volunteer students (*n* = 11) who filled in the questionnaire three times in 4 weeks. For COVID-19 prevalence and COVID-19 vaccination, the Cronbach Alpha internal consistency coefficients were 1.0.

### 2.3. Sample Size Relevancy

All 359 2nd–5th year dentistry students were addressed with an invitation to this study. The minimum number of study participants was calculated as 186. Formula (1) was used for the calculation using the online Netquest calculator [14]. (*N*) represents the study universe, i.e., all dentistry students in the 2nd-5th years, (*e*) represents a margin of error set at 5%, (*Z*) represents a confidence level at 95%, and (*p*) represents a standard heterogeneity at 50%.
(1)$$n=\frac{\;N\cdot Z^{2}\cdot p\cdot \left(1-p\right)}{\left(N-1\right)\cdot e^{2}+Z^{2}\cdot p\cdot \left(1-p\right)}$$

Formula (1). Relevant sample size calculation.

### 2.4. Data Collection

The link provided to study participants redirected them to an online questionnaire in Google Forms (Google, Mountain View, CA, USA). The questionnaire could be completed from 23 May 2022, to 23 June 2022, and answer data were stored in the Google Forms cloud database. On 24 June 2022, the dataset was downloaded.

### 2.5. Analysis

The data analysis was performed independently by three authors (Jan Schmidt, Lenka Vavrickova, and Martin Kapitan). Differences in analysis results were assessed and decided by the 4th author (Jakub Suchanek). Blank responses were not counted.

A person who received one dose of a single-dose vaccine or two doses of a two-dose vaccine was considered fully vaccinated. First and second vaccine boosters are considered the first and second vaccine doses received after the full vaccination has been completed. Full vaccination dose means the second dose of the two-dose vaccine or the first dose of the single-dose vaccine. The first case of COVID-19 infection is referred to as infection or primoinfection, the second case of infection is referred to as reinfection, and the third case of infection is referred to as the second reinfection.

The Czech general population data were mined from the Czech Ministry of Health database [15]. For COVID-19 infections, the data for age groups 18–24 and 25–29 were included in the analysis. For COVID-19 vaccination, the data for age groups 20–24 and 25–29 were included in the analysis. Age sorting of datasets and infection verification were determined by the current methodology of the Czech Ministry of Health [15,16,17]. Since there is an uneven representation of males and females within the study population, the general population data were approximated so that its sex ratio was identical to that of the study population to better reflect possible differences in prevalence and vaccination between sexes.

The data were sorted and analyzed in Microsoft Office Excel (version 2106 for Windows, Microsoft Corporation, Redmond, WA, USA) and GraphPad Prism (version 8.0.0 for Windows, GraphPad Software, San Diego, CA, USA). Statistical tests used for each calculation are listed in the Results chapter. Mann-Whittney test was used for statistical analysis of the incapacity period. The chi-square test with Yates’ correction was used for other statistical analyses of differences, and Babtista-Pike method was used for odds ratio (OR) and 95% confidence interval (CI).

## 3. Results

### 3.1. Response Rate and Center Distribution

All 359 students from the 2nd to 5th year were addressed with an invitation to this study. Of them, 240 participated in this study, representing a 66.9% response rate. No response was blank. Overall, 140 (58.3%) participants were students of the Faculty of Medicine in Hradec Kralove, and 100 (41.7%) participants were students of the Faculty of Medicine in Pilsen. The study results are statistically relevant as the minimum required number of participants, i.e., 186, was exceeded (Figure 1).

### 3.2. Sex and Age Distribution

The sex, age, and study year distribution of participants are illustrated in Figure 1.

### 3.3. COVID-19 Infection

Respondents were asked how many times they were infected with COVID-19, when the infections occurred, and how it was diagnosed. Additionally, they were asked about the place of infection, place of treatment, long-term complications, duration, and the time during which the illness prevented them from attending school.

#### 3.3.1. COVID-19 Prevalence

Overall, 120 (50.0%) respondents stated that they were infected with COVID-19. Out of them, 24 (10.0%) twice. The prevalence was 52.4% among females and 49.2% among males (*p* = 0.77). The prevalence among study years was as follows: 2nd year 53.8%, 3rd year 58.3%, 4th year 46.3%, 5th year 45.0%. Among 58 (24.2%) respondents who had no contact with patients during the pandemic, the prevalence was 58.6%. Out of 182 (75.8%) participants who worked with patients, the prevalence was 46.2%.

#### 3.3.2. COVID-19 Prevalence Based on the Vaccination Status

Out of 15 respondents who stated they were without full vaccination status, three (20%) were not infected with COVID-19, 12 (80%) were infected with COVID-19 at least once, and five (33.3%) were infected twice. Out of 94 respondents who stated they were fully vaccinated, 44 (46.8%) were not infected with COVID-19, 50 (53.2%) were infected with COVID-19 at least once, and 10 (10.6%) were infected twice. Out of 131 respondents who stated they received a booster dose, 73 (55.7%) were not infected with COVID-19, 58 (44.3%) were infected with COVID-19 at least once, and nine (6.9%) were infected twice.

For primoinfections, the difference between respondents who were not fully vaccinated and those who were fully vaccinated is not statistically significant. The difference between respondents who were not fully vaccinated and those who received a booster was statistically significant (*p* = 0.02, OR 5.0, 95% CI 1.5–17.0). The difference between respondents with full vaccination and booster was not statistically significant.

For reinfections, the difference between respondents who were not fully vaccinated and those who were fully vaccinated is statistically significant (*p* = 0.049, OR 4.2, 95% CI 1.3–14). The difference between respondents who were not fully vaccinated and those who received a booster was statistically significant (*p* = 0.0046, OR 6.8, 95% CI 2.0–23). The difference between respondents with full vaccination and booster was not statistically significant.

#### 3.3.3. COVID-19 Test-Verified Prevalence

As most statistics include only PCR and/or Antigen (PCR/Ag) verified or PCR-verified COVID-19 infection, such sorting is necessary for comparability. Out of 120 respondents who admitted being infected with COVID-19, a total of 94 (78.3%) replied that the infection was confirmed with some test, such as PCR, antigen, or antibody test. In 26 cases (21.7%), the diagnosis was based solely on clinical symptoms. A total of 92 (76.7%) of all primoinfections were PCR/Ag-verified representing 38.3% PCR/Ag-verified prevalence among all respondents. Reinfection was PCR/Ag-verified in 17 cases, representing 70.1% of all reinfections and a 7.1% PCR/Ag-verified reinfection rate among all respondents. The PCR test confirmed the infection in 68 cases, representing 56.7% of all infected respondents and 28.3% PCR-verified prevalence among all respondents. Reinfection was PCR-verified in 10 cases, representing 41.7% of all reinfections and a 4.2% PCR-verified reinfection rate among all respondents (Figure 2).

#### 3.3.4. Place of Infection

Out of 120 respondents who admitted being infected with COVID-19, a total of 61 (50.8%) were aware of where they were infected. Among them, 55 (90.2%) respondents stated they were infected in the out-of-school environment and six (9.8%) in the school environment.

#### 3.3.5. Hospitalization Rate

No participants were hospitalized due to COVID-19 infection.

#### 3.3.6. Incapacity Period

The median incapacity period, e.g., the time from the onset of the disease to the return to normal life, was 10 days (on average 10 days). The reasons were as follows: Quarantine (88 answers, 73.3%), Medical conditions (31 answers, 25.8%), and Other (1, 0.8%).

The median incapacity period among infected individuals after achieving full vaccination was seven days (on average eight days). Among individuals who were infected before achieving full vaccination, it was 11 days (on average 10 days). This difference is statistically significant (*p* = 0.0007, Mann-Whittney test).

#### 3.3.7. Long-Term Complications

Among respondents infected with COVID-19, 35 (29.2%) admitted long-term complications. The complications and their frequencies expressed as a total number of participants and percent of all infected participants were as follows: Feeling weak, inefficient (19, 15.8%); smell change (16, 13.3%); respiratory difficulties (13, 10.8%); taste change (13, 10.8%); mood swings, depression (10, 8.3%); headache (eight, 6.7%); muscle or joint pain (five, 4.2%); indigestion (four, 3.3%); dislike of food (two, 1.7%); increased body temperature (two, 1.7%). The average duration of complications was 63 days, and the median duration was 45 days.

#### 3.3.8. Protective Measures Used during Work with Patients

Out of all respondents, 58 (24.2%) respondents had no contact with patients during the epidemic. The remaining 182 (75.8%) had contact with patients, of which 182 (100%) used an FFP2 respirator, 137 (75.3%) used face barriers, such as a face shield (114, 62.6%) or protective glasses (71, 39.0%), and 66 (36.3%) protective clothing. A total of 183 respondents (76.25%) want to maintain protective measures even if they are not mandatory in the future, 18 (7.5%) do not, and 39 (16.25%) did not answer or provided an unclear answer. (Figure 3).

### 3.4. COVID-19 Vaccination

#### 3.4.1. Vaccination Rate

Overall, 15 participants (6.3%) received no vaccine dose, six (2.5%) participants received one vaccine dose, 88 participants (36.7%) received two vaccine doses, 130 (54.2%) participants received three vaccine doses, and one participant (0.4%) received four vaccine doses. A total of 225 (93.8%) respondents were fully vaccinated, 1st booster received 131 (54.6%) participants, and 2nd booster received 2 (0.8%) participants (Figure 4).

#### 3.4.2. Vaccine Type Used

Out of all 576 vaccine doses administered, the types used were as follows: Comirnaty (Pfizer/BioNTech) 507 (88.0%), Spikevax (Moderna) 42 (7.3%), Vaxzevria (AstraZeneca) 17 (3.0%), Janssen (Johnson & Johnson) eight (1.4%), and Sinopharm COVID-19 (Sinopharm) two (0.3%).

#### 3.4.3. Vaccination Adverse Effects

Out of all 576 vaccine doses administered, 191 administrations (33.2%) were associated with adverse effects, and 385 (66.8%) were not. Of all doses, six were the first dose of one dose vaccine, 219 were the first dose of a two-dose vaccine, 219 were the second dose of a two-dose vaccine, and 131 were a booster dose. The rates of complications and frequency of complication types are provided in Figure 5.

A higher COVID-19 infection rate was reported among individuals who developed some type of complication after receiving a booster dose compared to individuals who developed no complication after receiving a booster dose (*p* = 0.04, OR 2.3, 95% CI 1.1–4.9). No significant difference in COVID-19 infection rate was observed based on complications occurring after the first dose of the two-dose vaccine or the full vaccination dose.

#### 3.4.4. Reasons Leading to Avoiding COVID-19 Vaccination

Out of 15 respondents who were not vaccinated, 11 (73.3%) stated they refuse COVID-19 vaccination, two (13.3%) stated medical reasons for avoiding COVID-19 vaccination, one (6.7%) responded stated having a sufficient level of COVID-19 antibodies, and one (6.7%) respondent refused to answer.

#### 3.4.5. COVID-19 Vaccination Penetration

Out of 225 respondents who achieved full vaccination, 64 were infected with COVID-19 after achieving full vaccination, representing 28.4% vaccination penetration.

#### 3.4.6. Future Approach to COVID-19 Vaccination

All participants replied to this question. Overall, 62.1% of respondents have unconditionally or conditionally positive attitudes to future vaccination against COVID-19. 22.9% do not agree with future vaccination against COVID-19 (Figure 6).

#### 3.4.7. Approach to Mandatory Vaccination for Healthcare Professionals and the General Population

All respondents replied to this question. The majority of respondents agree with mandatory COVID-19 vaccination for healthcare professionals and the general population. Support is higher for mandatory vaccination of health workers (75.4%) than for the general population (62.9%). The negative opinion is held by 17.9% and 28.7%, respectively (Figure 7).

### 3.5. Comparison to Czech General Population Counterparts

#### 3.5.1. Test-Verified COVID Prevalence

PCR/Ag-verified COVID-19 prevalence among study participants at the end of the monitored period was 37.1%. In the Czech general population counterparts, it was 45.1%. The difference between the general Czech population counterparts and the study population is statistically significant: *p* = 0.015, OR 1.2, 95% CI 1.0–1.5.

PCR/Ag-verified COVID-19 reinfection among study participants at the end of the monitored period was 7.1%. In the Czech general population counterparts, it was 3.4%. The difference between the general Czech population and the study population was statistically significant: *p* = 0.003, OR 0.46, 95% CI 0.28–0. 76. A total of 76.5% of these reinfected students avoided a booster vaccine dose. The timeline of COVID-19 PCR/Ag-verified infections and reinfections is illustrated in Figure 8.

#### 3.5.2. COVID-19 Vaccination Rate

The COVID-19 full vaccination rate among study participants at the end of the monitored period was 93.8%. In the Czech general population, it was 67.1%. The difference between the Czech general population and the study population is statistically significant: *p* < 0.0001, OR 7.4, 95% CI 4.4–12.

The COVID-19 1st booster rate among study participants at the end of the monitored period was 54.6%. In the Czech general population, it was 24.4%. The difference between the Czech general population and the study population is statistically significant: *p* < 0.0001, OR 3.7, 95% CI 2.9–4.8. The timeline of COVID-19 full vaccination and 1st booster rates is illustrated in Figure 9.

## 4. Discussion

Medical education has been reduced during the COVID-19 pandemic. This policy affects the quality and availability of health care and should be retrospectively evaluated as to whether it was necessary. One of the possible evaluation parameters is the risk of infection during clinical teaching. For these purposes, it is possible to compare COVID-19 infection rates among students who attended clinical classes during the pandemic years and compare them with the infection rates of their general population counterparts. Such a comparison will indicate whether the higher risk of infection resulting from clinical work had an impact on COVID-19 prevalence among students.

We asked respondents where they were infected with COVID-19. The school environment was identified as a place of infection by 9.8% of respondents who were aware of where they were infected. However, this outcome is not reliable enough on its own, as it only represents the opinions of the respondents. A robust tool to assess the infection risk is the prevalence of COVID-19. Overall, 50% of respondents said they were infected with COVID-19. The infection rate was significantly lower among individuals who received a booster dose compared to those who were not vaccinated. Individuals with full vaccination or with a booster dose reported significantly lower reinfection rates compared to respondents who were not vaccinated. Respondents’ answers about whether or not they were infected with COVID-19 also include non-objective methods of diagnosis, e.g., based only on symptoms. Additionally, it is not comparable with the official statistics of COVID-19 prevalence, which are based only on test-verified infections. In the past, only PCR tests were used to verify COVID-19 infection in most countries. The same was the case in the Czech Republic. However, with the change in the availability and quality of antigen tests, they were also included in the verification methodology. According to the Czech Ministry of health, the methodology for COVID-19 verification includes PCR and/or an antigen test [17]. Thus, to obtain objective and comparable data, it is necessary to count only the prevalence data verified with a PCR and/or an antigen test. Among the respondents, the infection was confirmed by PCR/Ag test in 37.1%. A comparison of COVID-19 prevalence between the respondents and their peers from the general Czech population (45.1%) revealed that the study participants were statistically significantly less affected by COVID-19. Such a result is crucial because it shows that despite the high risk to which dental students were exposed during clinical education, it did not lead to higher COVID-19 morbidity.

This result could be due to several factors. The prevalence curve was similar in both groups until January 2021, when infections among students began to decrease (Figure 8). At the beginning of 2021, the Czech Republic launched the COVID-19 vaccination, which prioritized healthcare professionals. In our previous studies, we demonstrated the effect of priority COVID-19 vaccination among healthcare workers as a significant decrease in COVID-19 infection compared to the general population [18,19,20]. A similar effect can be assumed in the case of students as well. Although the student vaccination rate was low until May 2021 (11.3%), it was still almost three times higher than among the general population of a similar age (Figure 9). In the rest of 2021, full vaccination among students rose to over 90%. It was approximately a third higher than in the general population, although, at that time, the accessibility of vaccination was the same for all individuals in the Czech Republic. This demonstrates that students not only had earlier access to vaccination, but overall acceptance of vaccination was higher than in their general population peers. The difference in acceptance of the 1st booster was even higher. Application of the 1st booster was more than doubled among students compared to the general population, which could be the reason for the significantly lower increase in infections among students during the beginning of 2022. These differences can be considered essential aspects of reducing COVID-19 prevalence. A further protective factor could have been the high rate of protective equipment usage, especially FFP2 respirators and shields. All students used FFP2 respirators and three-quarters a face barrier when working with patients. The protective effect of these protective measures against the infection of COVID-19 has been proven [21,22]. High compliance in the use of protective measures contributed to the reduction of infection among the respondents. The vaccination protective effect can also be demonstrated in the vaccination rate of reinfected individuals. PCR/Ag-verified reinfection rate was significantly higher among students. However, 76.5% of these reinfected students avoided booster vaccination dose. Another possible reason why the reinfection rate was higher among students is that compliance with self-reporting of infections in the general public decreased during the pandemic [23]. On the other hand, dental students could maintain higher compliance due to their education.

Interestingly, the symmetry of the students’ and their peers’ prevalence curves was disturbed at the beginning of the year 2021 and also 2022. The first split is visible in the period from January to March 2021 and the second from January to March 2022, both resulting in amplifying the prevalence gap between students and their general population counterparts. These changes correlate with the difference in full vaccination and 1st booster application. Another possible contributing factor is that June and February are the months of the exam period. Students are often isolated from society due to the high workload of exam preparations. Other factors that could have influenced the increase in infection and reinfection rates in the first quarter of 2022 could be the waning protection of applied vaccines, insufficient revaccination, new variants of the SARS-CoV-2 virus, or the influence of the season.

As this study is the first to compare the pandemic’s impact on medical students and their peers, comparisons with other studies are limited. Many studies present hesitancy, acceptance, or views of medical students on vaccination, but only a small number of studies provide data on real vaccination rates [24,25,26,27,28,29]. Studies investigating the willingness of medical students to get vaccinated against COVID-19 or reporting on their vaccination status showed high compliance among respondents. The vaccination rates vary based on the place and time when the analysis was performed. In February 2021, an extensive study focused on the global prevalence and drivers of dental students’ COVID-19 vaccine hesitancy was conducted. 6639 students from 22 countries participated in it. The authors concluded that the overall acceptance level (63.5%) of dental students for COVID-19 vaccines worldwide was suboptimal [30]. These findings are consistent with the results of a 2020 study of US dental students. This study demonstrated significant doubts among students about safety, as only 65.6% of respondents reported trusting the information they received about the COVID-19 vaccine from the healthcare authorities, 67.4% were willing to recommend the vaccine to their relatives and friends, and 68.6% would recommend it to their patients [31]. However, the data show that students’ willingness to get vaccinated improves over time. A study among Romanian medical students performed in January–March 2021 showed that 88.5% had a pro-vaccination attitude [32]. A study by Kateeb performed during February and March of 2021 assessed the predictors related to the willingness of 417 Palestinian dental students to receive the COVID-19 vaccine when it becomes available. 57.8% were willing to take the COVID-19 vaccine when it became available to them, 27% were hesitant, and 14.9% were not willing to get vaccinated [33]. A study by Talarek et al. published in June 2021 aimed to determine influenza vaccine uptake among 675 students of the Medical University of Warsaw, Poland, and their intention to receive a hypothetical COVID-19 vaccine. 94.6% of students expressed their intention to receive a COVID-19 vaccine. In January 2021, Saied et al. described COVID-19 vaccination hesitancy, beliefs, and barriers among 2133 Egyptian medical students [27]. 90.5% considered COVID-19 vaccines important, 46% had vaccination hesitancy, and an equal percentage (6%) either definitely accepted or refused the vaccine. A March 2021 survey study among 888 medical students in Kazakhstan showed that only 2% had been vaccinated against COVID-19 [34]. In July and August 2021, Peterson et al. conducted a survey study among 234 students at the Texas Tech University Health Sciences Center in Lubbock, showing that the vaccination rate was 91.8% [35]. This result is very similar to the rate in our study. Another survey study performed between April and May 2021 among 496 Dokkyo Medical University, Japan, students revealed that 89.1% received a second COVID-19 vaccine dose [36]. In September 2021, Bader et al. published a survey study performed among 204 students of the College of Medicine at King Saud bin Abdulaziz University, Saudi Arabia, showing that 66% of them received a vaccine against COVID-19. In March 2021, Sovicova et al. conducted a study reporting on the vaccination status of Slovak medical students [37]. Out of 1228 respondents, 71.7% were vaccinated, and 28.3% were not. In summary, results vary by time, place, and methods used, and the vaccination rates are influenced by local conditions such as vaccine availability or socioeconomic conditions of the country. Since the studies do not provide data on the vaccination of the respondents together with the vaccination rates of their peers from the general population, it is not possible to perform their comparison.

Studies on the prevalence of test-verified COVID-19 among medical students are even less available. Amjad et al. surveyed via online questionnaire 1830 students of the University of Jordan between December 2020 and February 2021. PCR-verified COVID-19 prevalence was 13% [38]. Silva et al. conducted a multicentric study on COVID-19 prevalence among Brazilian healthcare workers and students during August and December 2020 [39]. Out of 564 students involved, 11% reported PCR-verified COVID-19 infection. None of these studies provide a comparison of data with peers from the general population.

In ideal model conditions, 100% of study participants would be in contact with patients during the pandemic. However, such a state is practically difficult to achieve. In our study, 75% of respondents participated in clinical work during the pandemic, which is a sufficient proportion for the higher infection risk to be reflected in the overall results. Furthermore, some students were compulsorily called to help in overburdened hospitals at a time of restrictions on clinical teaching. They usually performed non-medical work, such as sanitation, which is also associated with a high risk of infection transmission. Thus, even these students were exposed to an increased risk of infection. However, data on how many of them and the type of work performed are not available. Furthermore, it would be ideal for the comparative group (general population) to be approximated according to the geographical location of the students as there as there may be differences between urban and rural populations. However, further details about the students’ residence combined with age and sex could violate anonymity because the number of dentistry students is small. Additionally, in the Czech Republic, many students have multiple places of residence and live part of the week in the place of study and part in the place where they come from. Therefore, determining the geographical characteristics of the participants could be misleading. Another limitation is the number of participants. A larger number of study participants would be ideal; however, the dentistry study program is attended by only a few dozen individuals each year. The total number of 240 participants representing a response rate of 66.9%, exceeds the minimum relevant sample of 186 respondents. Compared to similar studies, the number of respondents in our work can be considered sufficient. Additionally, the population of individual countries should be considered as well. The Czech Republic is a small country in central Europe with a population of 10 million. In terms of the participants/population ratio, the number of participants in our study is above average. Furthermore, our study is the first to combine vaccination and PCR/Ag-verified prevalence data of medical students and its comparison to data of general population peers. This comparative model is unique and provides more informative data than a simple expression of vaccination and prevalence rates.

## 5. Conclusions

Dental students showed high compliance with the use of protective equipment and COVID-19 vaccination. When in contact with patients, 100% of students used an FFP2 respirator, and 75% used a face barrier. Compared to counterparts from the general population, dental students’ full vaccination and 1st booster rates were significantly higher. The combination of extensive protective measures and high vaccination against COVID-19 led to significantly lower COVID-19 prevalence among the students compared to their general population counterparts.

## Figures and Tables

**Figure 1 vaccines-10-01927-f001:**
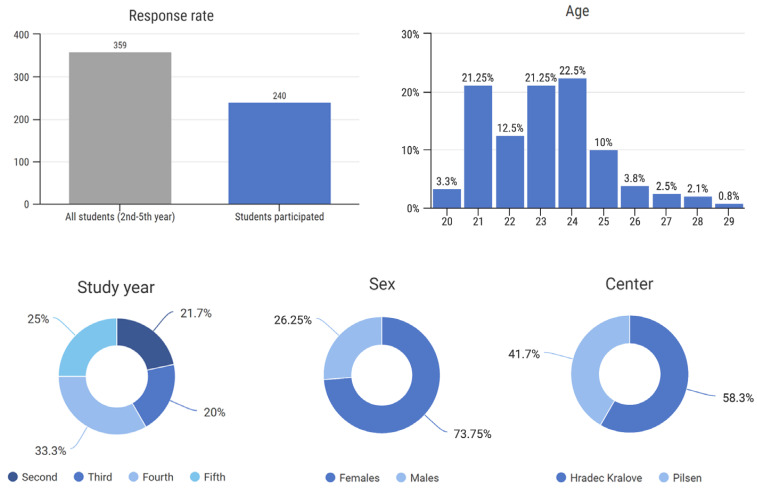
Response rate and center, age, sex, and study year distribution of participants.

**Figure 2 vaccines-10-01927-f002:**
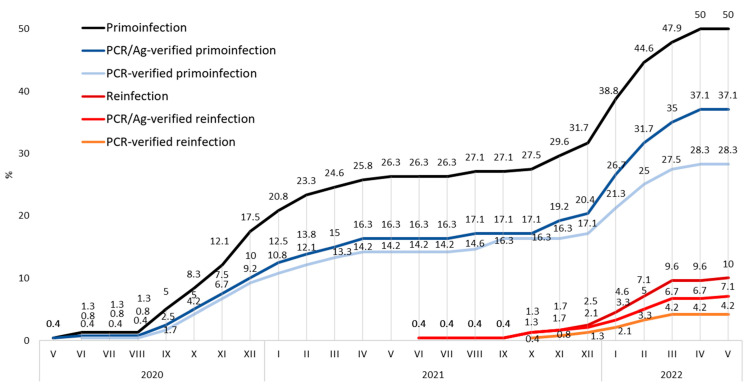
Primoinfection and reinfection rates among participants.

**Figure 3 vaccines-10-01927-f003:**
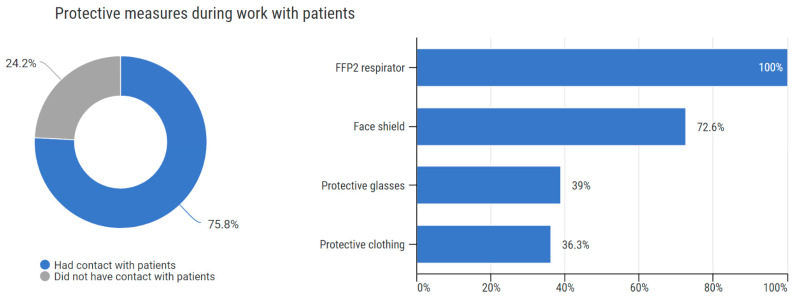
Protective measures during work with patients.

**Figure 4 vaccines-10-01927-f004:**
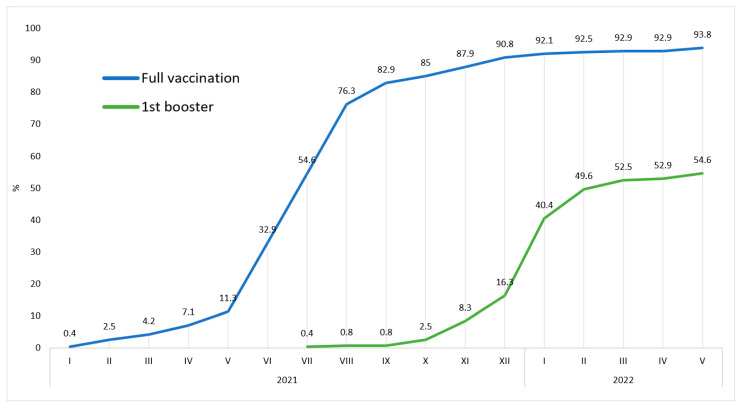
The vaccination rate among participants.

**Figure 5 vaccines-10-01927-f005:**
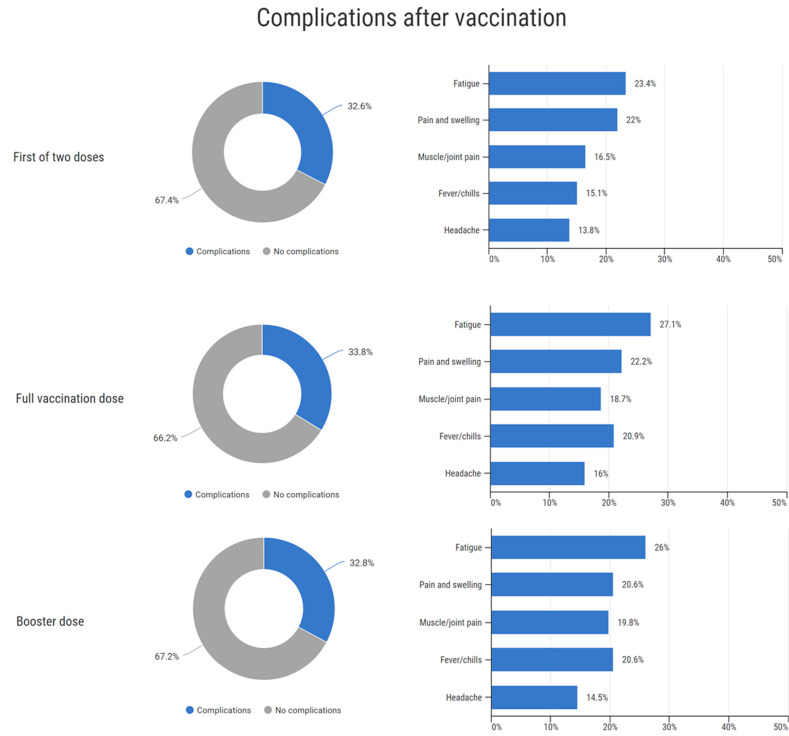
Complications after vaccination.

**Figure 6 vaccines-10-01927-f006:**
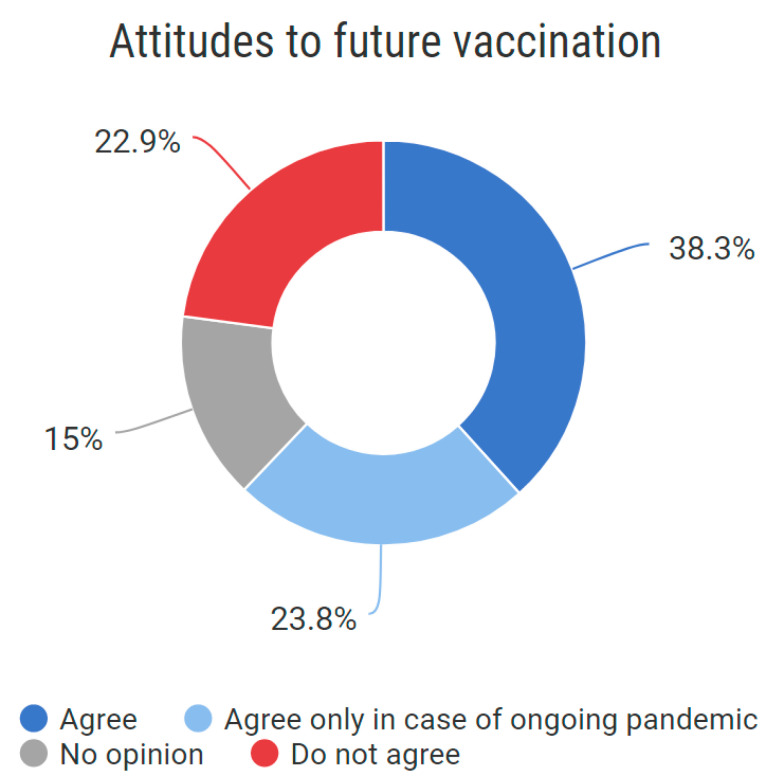
Attitudes to future COVID-19 vaccination.

**Figure 7 vaccines-10-01927-f007:**
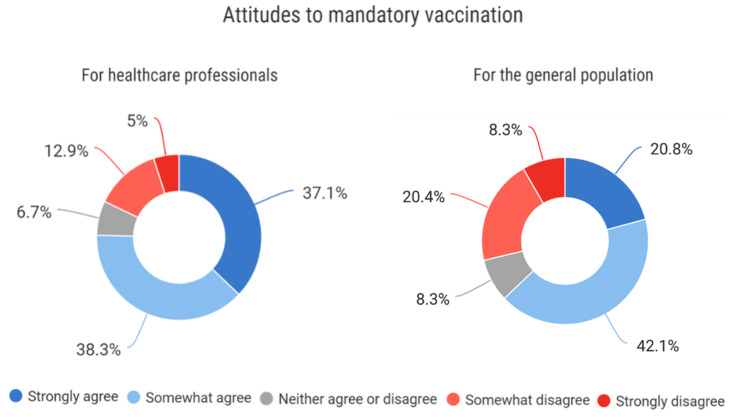
Attitudes to mandatory vaccination. The percentages may not add up to 100% as numbers are rounded.

**Figure 8 vaccines-10-01927-f008:**
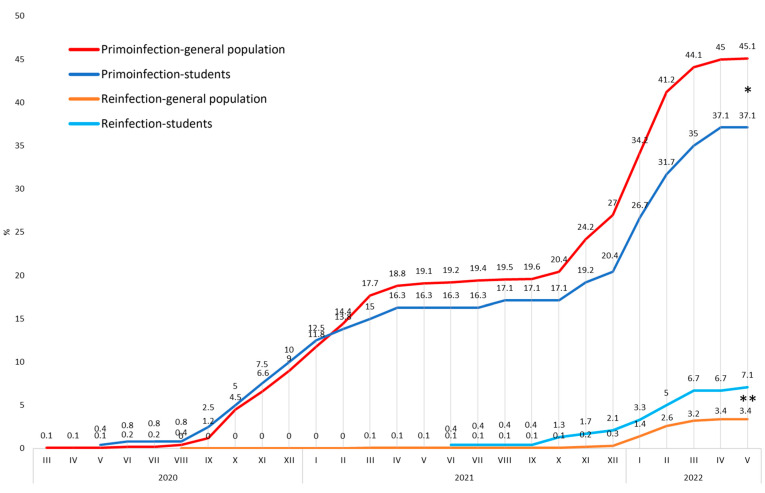
Comparison of primoinfection and reinfection rates among study participants and their peers from the Czech general population. * indicates *p* < 0.05, ** indicate *p* < 0.01.

**Figure 9 vaccines-10-01927-f009:**
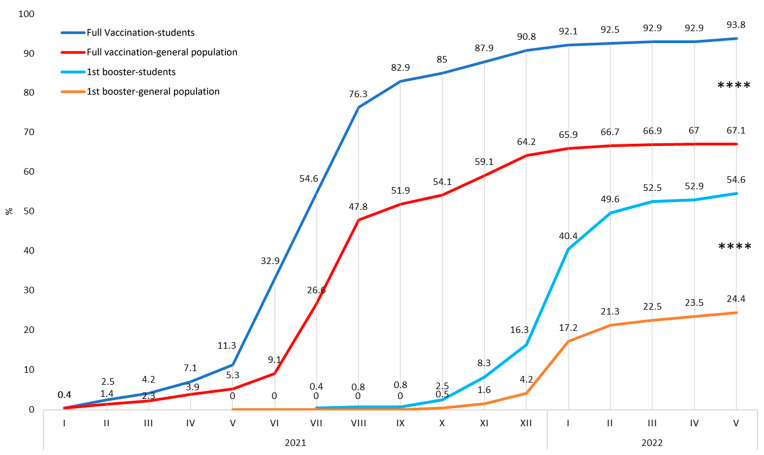
Comparison of vaccination rates among Czech dentistry students and their peers from the Czech general population. **** indicate *p* < 0.0001.

## Data Availability

The dataset is available on demand from the corresponding author.

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
