# Peer review of "COVID-19 among Czech Dentistry Students: Higher Vaccination and Lower Prevalence Compared to General Population Counterparts"

_vaccines, 2022, doi:10.3390/vaccines10111927_

Round 1

Reviewer 1 Report

Please describe the respirators including manufacturer. In many countries, N95 and KN95 masks are used.

A person who received one dose of a single-dose vaccine or two doses of a two-dose 114

vaccine was considered fully vaccinated.

Comment: What about the fact that protection wanes after a few months?

Thus, the increase in infection as well as reinfections in 2022 could be due to waning protection from the original vaccines, lower booster rates, new COVID-19 variants, and reduced protection due to the sun in summer related to vitamin D, nitric oxide, and higher temperature. However, it is interesting that reinfection rates were higher for dental students than their peers? Can a hypothesis be proposed to explain that?

By the end of May 2022, COVID-19 full vaccination and 1st booster 26
rates among students were 93.8% and 54.6%

Comment: Can comparisons of COVID-19 infections be made for those with or without full vaccination as well as for 1st booster?

3.4.3. Vaccination adverse effects

221

Out of all 576 vaccine doses administered, 191 administrations (33.2%) were associ- 222
ated with adverse effects, and 385 (66.8%) were not. Of all doses, 225 were first, 219 were 223
second, and 131 were third. The rates of complications and frequency of complication 224
types are provided in Figure 7.

Comment: How much time was lost due to complications?

Also, what is the correlation between adverse effects of vaccination/booster and developing COVID-19?

Better immune systems, such as due to higher vitamin D levels or better diet reduce risk of COVID-19 and may reduce risk of adverse effects of vaccination.

Comment: It might be discussed in the manuscript that the lower COVID-19 rate among dental students could be due to using respirators, face shields, etc., not just vaccinations.

Regarding medical student vaccination rates elsewhere, a search of scholar.google.com finds a number of other publications. Suggest citing some of them and, perhaps, making a table showing vaccination rates for students based on these publications and the ones cited already in the manuscript.

Romanian medical students' attitude towards and perceived knowledge on COVID-19 vaccination

A Bălan, I Bejan, S Bonciu, CE Eni, S Ruță - Vaccines, 2021 - mdpi.com

… has weighed down and will continue to do so heavily on the healthcare system as long
as the 
vaccination rate in the general population remains low. Enthusiastic and dynamic …

Odds Ratio Estimation of Medical Students' Attitudes towards COVID-19 Vaccination

M Sovicova, J Zibolenova, V Svihrova… - International Journal of …, 2021 - mdpi.com

… attitudes of Slovak medical students to COVID-19 vaccination. A … It was performed in four
medical faculties with students in … with a lower COVID-19 vaccination rate. Surprisingly, routine …

Save Cite Cited by 21 Related articles All 12 versions 

[HTML] nih.gov

Vaccine hesitancy: Beliefs and barriers associated with COVID19 vaccination among Egyptian medical students

SM Saied, EM SaiedIA Kabbash… - Journal of medical …, 2021 - Wiley Online Library

… COVID-19 vaccine acceptance among medical students, the current study was formulated
targeting 
medical students … South Carolina college students reported vaccination costs as one …

COVID-19 vaccination hesitancy among health care workers, communication, and policy-making

SM Toth-Manikowski, ES SwirskyR Gandhi… - American journal of …, 2022 - Elsevier

… and COVID-19 Vaccine Attitudes Scale items with statistical significance at predicting COVID-19
vaccination… One method supported by our data to improve vaccination rates is to include …

Save Cite Cited by 34 Related articles All 8 versions

[PDF] nejm.org

Association between covid-19 vaccination and influenza vaccination rates

RK LeuchterNJ JacksonJN Mafi… - … Journal of Medicine, 2022 - Mass Medical Soc

… low Covid-19 vaccination rates would be associated with decreases in influenza vaccination
rates. … in influenza vaccine uptake at the state-population level during the pandemic after …

Save Cite Cited by 6 All 7 versions

[PDF] mdpi.com

Influenza vaccination coverage and intention to receive hypothetical ebola and COVID-19 vaccines among medical students

E Talarek, J Warzecha, M Banasiuk, A Banaszkiewicz - Vaccines, 2021 - mdpi.com

… vaccination against COVID-19 may be one of the reasons underlying the increased influenza
vaccination rate among medical students … symptoms and COVID-19 symptoms are similar, 

Significant digits. The general rule is that no more non-zero digits should be given than are justified by the uncertainty of the value.

See "Too many digits: the presentation of numerical data"

https://www.ncbi.nlm.nih.gov/pmc/articles/PMC4483789/

If the uncertainty is greater than about 7%, only two non-zero digits are justified.

P values should be given to two decimal places unless the first two are 00 or the number lies between 0.045 and 0.054.

Thus

OR 7.318, 95% CI 4.339-12.340; p < 0.0001, OR 3.856, 95% CI 2.991-4.972

Should be

OR 7.3, 95% CI 4.3-12.3; p < 0.0001, OR 3.9, 95% CI 3.0-5.0

Please review all numbers in abstract, text, tables, and figures and adjust accordingly.

Reviewer 2 Report

Clinical education is perceived to be associated with a  high risk of Covid-19 pandemic spread.   The aim of the study was to provide data on COVID-19 prevalence and vaccination of  Czech dental students and its comparison with related data of their Czech general population counterparts. The study assessed the hypothesis  as to whether medical education held during the pandemic was associated with an increase in COVID-19 prevalence among Czech dentistry students.

The manuscript is well written and has a clear friendly structure (Introduction, Materials and Methods, Results,  Study Limitations, Discussion and Conclusions). The subject is  interesting and very topical,  as the paper deals with  the Covid-19 prevalence and degree of vaccination among Czech dentistry students. The background is transparent and informative. The materials and methods  are clearly and thoroughly  described. The discussion is sufficient and   is supplemented with limitation section.  The paper has 25 adequate references.  The text is completed by 11 relevant figures.

However, there are a few points that need to be completed / clarified:

1. All statistical tests used in the analysis  should be described in details in  the subsection 2.4 Analysis (not  mention for the first time in the Results chapter).

2. In subsections 3.4.6 and 3.4.7  there are only two figures ( Figure 8 and Figure 9) with no  further explanation.  These sections should be supplemented with at least a short description /comments to the most important results presented in the form of figures.

3. With regard to Figure 9, it is also unclear what data they come from? From the study or more likely from the database of the Czech Ministry of Health? It must be clearly stated.

4. The section ‘Study limitations’ should be moved  after the  ‘Discussion’ section.

Reviewer 3 Report

The study showed that dentals students form the Faculty of Medicine in Hradec Kralove and the Faculty of Medicine in Pilsen (Check Republic) had significantly higher vaccination rate and lower COVID-19 prevalence (p=0.03) comparing with the general Check population of the same age. Infection was confirmed by PCR and/or antigen test. There is a lack of data on the test-verified prevalence of COVID-19 among medical students. Therefore, current investigation is relevant.

Remarks

I have doubts on the choice of control sample. The data from general population were imported based on the age, while other factors were not considered. In the medical student sample 74% were females, and 26% males, which shows a large deviation in the gender ratio, therefore, gender had to be taken into account when composing the control sample. Also, “The prevalence was 52.4% among females and 49.2% among males” (it is not clear if this difference is statistically significant)

Authors did not describe the COVID-19 prevalence and vaccination differences in Check Republic based on geographic regions, cities/towns. It is not clear whether these differences are important and therefore should be considered by creating control samples. Previous studies in different countries have shown infection prevalence and vaccination difference depending on education, population density, etc

Abstract: the main reasons explaining results are higher vaccination, which is prioritized for medical students (also it is obvious that medical students are more positive about vaccination)

and high rate of protective equipment used by dental students, i.e., respiratory, shields, protective glasses and clothes

why authors did not choice to compare vaccination and infection prevalence rate between dental students and other students of the same universities (belonging not to medical sciences and natural sciences)?

Other remarks:

L72-73 must be explained in more detail, according to what? according to the total proportion of infected persons, mortality or other factors

L136 change to ….was exceeded (Figure 1)…same to Figure 5, FIigure 6, etc.

FIGURE 1 there is a lack of data about study years. Prevalence was also compared according to the study years.

Figure 3 is redundant, do not provide additional necessary information; also, this is rather subjective data

Figure 4 also unnecessary, it is clear from the text 

Figure 7, not clear differentiation, since some vaccines are on-dose vaccines and others two-dose vaccines.

L229-232 from 1 to 9, numbers should be written in words

From Results ” Chi-square test with Yates’ correction, Babtista-Pike method“ move to Methods section

L394-396 Conclusions is incomplete, and the focus is somewhat misleading. “The results show that the high risk of COVID-19 infection as asociated with clinical teaching did not lead to higher COVID-19 prevalence among dental students.” This is caused by extensive security measures used by dental students (Figure 5) and higher vaccination rate (figure 11). Please, broaden conclusions.
